# Towards Stable and Robust AdderNets

**Minjing Dong**[1,2], **Yunhe Wang**[2]*, **Xinghao Chen**[2], **Chang Xu**[1]
[1]School of Computer Science, University of Sydney
[2]Huawei Noah's Ark Lab
mdon0736@uni.sydney.edu.au, yunhe.wang@huawei.com,
xinghao.chen@huawei.com, c.xu@sydney.edu.au

## Abstract

Adder neural network (AdderNet) replaces the original convolutions with massive multiplications by cheap additions while achieving comparable performance thus yields a series of energy-efficient neural networks. Compared with convolutional neural networks (CNNs), the training of AdderNets is much more sophisticated including several techniques for adjusting gradient and batch normalization. In addition, variances of both weights and activations in resulting adder networks are very enormous which limits its performance and the potential for applying to other tasks. To enhance the stability and robustness of AdderNets, we first thoroughly analyze the variance estimation of weight parameters and output features of an arbitrary adder layer. Then, we develop a weight normalization scheme for adaptively optimizing the weight distribution of AdderNets during the training procedure, which can reduce the perturbation on running mean and variance in batch normalization layers. Meanwhile, the proposed weight normalization can also be utilized to enhance the adversarial robustness of resulting networks. Experiments conducted on several benchmarks demonstrate the superiority of the proposed approach for generating AdderNets with higher performance.

## 1 Introduction

By hint of the success of Deep Convolutional Neural Networks (DCNNs), a wide range of computer vision tasks can be tackled with satisfactory performance, including image classification [13, 9, 10, 25], super-resolution [32, 4, 12, 22], object detection [16, 19, 7, 8], *etc.*. DCNNs are mainly trained and inferred on graphics processing unit (GPU) devices which suffice to fulfill the high computational complexity and enormous energy consumption. However, it is intractable for DCNNs to be deployed on low-power devices due to the limited computational speed and power capacity, such as mobile and embedded devices. As a result, substantial research efforts have been devoted to energy reduction and speed acceleration of DCNNs [3, 29, 1, 26].

Recently, Chen *et al.*[1] advocate the use of $\ell_1$-distance for similarity measure instead of cross-correlation in CNNs to replace multiplications with additions. Compared with multiplications, addition operations are much cheap, which benefits the power-efficiency [28, 23, 30]. Adder neural network (ANN) has demonstrated extraordinary performance on several computer vision tasks with huge energy reduction, which can be seen as a good complement to the classical CNNs.

The performance achieved by ANNs on classification is impressive, but some observations raise our concerns. *e.g.*, the optimization of ANNs is not stable as CNNs where the test accuracy curve has dramatic fluctuations during the optimization, as shown in Figure 1 (a). The test accuracy of ANNs has a large variation until the end of training while that of convolution networks is much more stable. Furthermore, weights of ANNs have demonstrated a significantly different statistical

---

*Corresponding author.

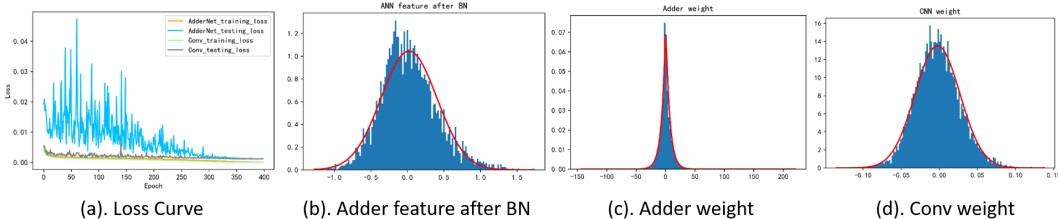

| (a). Loss Curve | (b). Adder feature after BN | (c). Adder weight | (d). Conv weight |

Figure 1: Observations of AdderNet. Training and testing loss curves of ANN and CNN in (a). Histograms over the AdderNet features after batch normalization layer follows Gaussian distribution in (b). Histograms over the AdderNet weight follow Laplace distribution with large variance while Conv weight follow Gaussian distribution with small variance in (c) and (d) respectively.

property from those of CNN weights. The histograms over ANN and CNN weights are shown in Figure 1 (c) and (d) respectively. The red curve in (c) denotes a Laplace distribution with a mean of $0.32$ and a variance of $88.47$ and the one in (d) denotes a normal distribution with a mean of $-0.002$ and a variance of $0.001$. Although the means of these two distributions are similar, there exists a serious discrepancy between their variances. Given this statistical property difference, those sophisticated training strategies previously developed for CNN might not fulfill their potential when straightforwardly applied to ANNs.

In this paper, we provide an exhaustive analysis of the variance in ANNs. Taking a one-layer adder forward as an example, we demonstrate the large variance of ANN weights can be the major cause of the instability of running mean and variance in batch normalization layer which vibrates the test accuracy. Weight normalization is therefore a natural idea to constrain the variance of weights. To preserve the representation capability of the normalized weights, trainable scaling and shift coefficients are further introduced to achieve an Adaptive Weight Normalization (AWN). By doing so, the variance of ANN weights can be normalized to acceptable levels which significantly stabilizes the batch normalization layers and makes pretrained ANN easily applied to other tasks. Notably, we identify a natural advantage of ANNs to be robust to adversarial perturbations, stemming from $\ell_1$-distance and the running mean of batch normalization layer in ANNs. The reduction of weight variation across channels further boosts the defense ability against attacks. Without adversarial training, AWN with ANN on ResNet-32 improves adversarial accuracy by $59.73\%$ compared with CNN under PGD attacks. AWN presents superior stability under a series of evaluations and outperforms vanilla ANN in detection task on VOC benchmark with 2.7 mAP improvement.

## 2 Preliminary

**Adder Neural Networks.** To significantly reduce energy costs, Chen *et al.*[1] proposed an Adder Neural Network to release the burden of multiplications in traditional convolution networks by replacing them with additions. Consider an intermediate layer in deep convolution neural network with weight $W \in \mathbb{R}^{d \times d \times c_{in} \times c_{out}}$ where $d$ denotes the kernel size and $c_{in}$ and $c_{out}$ are the number of input and output channel respectively. Given the input feature map $X \in \mathbb{R}^{H \times W \times c_{in}}$ where $H$ and $W$ are the height and width of input feature, the adder operation is defined as

$$Y(m, n, c) = -\sum_{i=1}^{d} \sum_{j=1}^{d} \sum_{k=1}^{c_{in}} |X(m+i, n+j, k) - W(i, j, k, c)|. \tag{1}$$

Although the adder operation enables ANNs to achieve similar performance in classification tasks with energy efficiency, the observations in Figure 1 shadow ANNs. The unstable test accuracy curve and large variance almost everywhere in ANNs arouse our interest in the analysis of ANN variance.

**Adversarial Attack.** Given the input $x \in \mathbb{R}^D$ and the annotated label $y \in \mathbb{R}^C$ where $D$ is the dimension of input and $C$ is the number of classes, the network $\mathcal{N}$ maps perturbed input $\tilde{x}$ to $\tilde{y} = \mathcal{N}(\tilde{x}; W)$ where $\tilde{x} = x + \delta$. The objective of adversarial attacks is to find the perturbed input which maximizes the classification loss as

$$\tilde{x} = \operatorname*{argmax}_{\tilde{x}:\|\tilde{x}-x\|_p \leqslant \epsilon} \ell(\mathcal{N}(\tilde{x}, W), y), \tag{2}$$

where the perturbation is constrained by its $l_p$-norm. Through variance study of ANNs and $\ell_1$-distance analysis, we demonstrate the potential adversarial robustness of ANNs can be activated through proposed inference strategy with AWN.

## 3 Variance Study of AdderNet

To analyze the variance of ANNs, we first consider a one-layer adder forward with batch normalization [11] and ReLU as activation function. Given output feature $y_{l-1}$ from previous layer, the forwarding of layer $l$ is formulated as

$$x_l = max(0, y_{l-1}), \; y'_l = -\sum_{}^{n} |x_l - W_l|, \; y_l = \gamma \frac{y'_l - \mu_l}{\sigma_l} + \beta, \tag{3}$$

where $\mu_l$ and $\sigma_l$ denote the mean and standard deviation, respectively, $W_l$ is the weight, $n$ represents the size of weight of one output channel $n = d \times d \times c_{in}$, and $\gamma$ and $\beta$ stand for the rescale and shift parameters.

Before we proceed to the weight variance analysis, we first make several assumptions that are empirically plausible. After the batch normalization, $y_{l-1}$ is supposed to follow a normal distribution $\mathcal{N}(0, \sigma^2)$. From empirically observation of a ResNet-18 with ANN on ImageNet, as shown in Figure 1 (b) and (c), adder feature after BN layer follows a distribution with mean of $0.02$ and variance of $0.38$ while the corresponding weight with mean of $0.32$ and variance of $88.47$, the adder weights follow Laplace distributions with a large variance but their mean is close to 0 after training. Thus, we assume $W_l$ follows $\mathcal{L}(\mu, b)$ with the mean as $\mu$ and the variance as $2b^2$ where $2b^2 \gg \sigma^2$.

Now we compute the mean and variance of each variable. The activation $x_l$ follows the Rectified Gaussian distribution. With the law of total expectation, the mean of $x_l$ forms the one-side truncation of lower tail $E[x_l|x_l > 0]$ which can be computed based on the property of Truncated normal distribution as

$$
\begin{aligned}
E[x_l] &= E[x_l|x_l > 0] \cdot P(x_l > 0) + E[x_l|x_l \leq 0] \cdot P(x_l \leq 0) \\
&= E[x_l|x_l > 0] \cdot P(x_l > 0) + 0 = [0 + \sigma \frac{\phi(0)}{1 - \Phi(0)}] \cdot \frac{1}{2} = \frac{\sigma}{\sqrt{2\pi}},
\end{aligned} \tag{4}
$$

where $\phi(\cdot)$ denotes the probability of standard normal distribution and $\Phi(\cdot)$ denotes the cumulative distribution function. Similarly, with the law of total expectation, the variance of $x_l$ is broken down into variants of expectation and the variance of one-side truncation of lower tail $Var(x_l|x_l > 0)$, which can be computed through Truncated normal distribution as

$$
\begin{aligned}
Var[x_l] &= E[x_l^2] - (E[x_l])^2 = -(E[x_l])^2 + E[x_l^2|x_l > 0] \cdot P(x_l > 0) \\
&= P(x_l > 0)[Var(x_l|x_l > 0) + (E[x_l|x_l > 0])^2] - (E[x_l])^2 \\
&= [\sigma^2(1 - (\frac{\phi(0)}{1 - \Phi(0)})^2) + \frac{4\sigma^2}{2\pi}] \cdot \frac{1}{2} - \frac{\sigma^2}{2\pi} = \sigma^2(\frac{1}{2} - \frac{1}{2\pi}).
\end{aligned} \tag{5}
$$

Since we assume $2b^2 \gg \sigma^2$ and the variance is further reduced after activation, the distribution of $x_l - W_l$ is overwhelmed by $W_l$ to form a Laplace distribution as $\mathcal{L}(\frac{\sigma}{\sqrt{2\pi}} - \mu, \sqrt{b^2 + \frac{(\pi-1)\sigma^2}{4\pi}})$. For simplicity, let $\tau_\mu = \frac{\sigma}{\sqrt{2\pi}} - \mu$ and $b + \tau_\sigma = \sqrt{b^2 + \frac{(\pi-1)\sigma^2}{4\pi}}$. Note that the standard deviation of output last layer $\sigma$ and the mean of weight $\mu$ are both close to zero, which makes $\tau_\mu$ and $\tau_\sigma$ small values. Although it is difficult to directly derive the closed-form distribution expression of $y'_l$, it can be approximated based on variable $x_l - W_l - \tau_\mu$. According to the properties of Laplace distribution, the absolute function of $\mathcal{L}(0, b)$ follows an Exponential distribution, with which $y'_l$ can be approximated as

$$y'_l = -\sum_{}^{n} |x_l - W_l| \geq -\sum_{}^{n} [|x_l - W_l - \tau_\mu| + |\tau_\mu|], \tag{6}$$
$$\text{where } |x_l - W_l - \tau_\mu| \sim \text{Exp}([b + \tau_\sigma]^{-1}),$$

where Exp denotes Exponential distribution. Based on the property of Exponential distribution, $E[|x_l - W_l|] = b + \tau_\sigma$ and $Var[|x_l - W_l|] = (b + \tau_\sigma)^2$. With Eq. 6, the lower boundary of $E[y'_l]$

can be derived as

$$E|x_l - W_l| \leq E|x_l - W_l - \tau_\mu| + |\tau_\mu|,$$
$$E[y_l'] \geq -n(b + \tau_\sigma + |\tau_\mu|). \tag{7}$$

With inequality in Eq. 7, the lower boundary of $Var[y_l']$ can be derived by the law of the unconscious statistician as

$$\begin{aligned}
Var[y_l'] &= \sum_{}^{n}[E[|x_l - W_l|^2] - (E[|x_l - W_l|])^2] \\
&= \sum_{}^{n}[Var(x_l - W_l) + (E(x_l - W_l))^2 - (E(|x_l - W_l|))^2] \\
&\geq n[Var(x_l - W_l) + (E(x_l - W_l))^2 - (E|x_l - W_l - \tau_\mu| \\
&\quad + |\tau_\mu|)^2] = n[2(b + \tau_\sigma)^2 + \tau_\mu^2 - (b + \tau_\sigma + |\tau_\mu|)^2] \\
&= n[(b + \tau_\sigma)^2 - 2|\tau_\mu|(b + \tau_\sigma)].
\end{aligned} \tag{8}$$

Taking the lower boundary of mean and variance of $y_l'$ in Eq. 7, 8, the batch normalization layer can be computed as

$$y_l = \gamma \frac{y_l' + n(b + \tau_\sigma + |\tau_\mu|)}{\sqrt{n[(b + \tau_\sigma)^2 - 2|\tau_\mu|(b + \tau_\sigma)]}} + \beta. \tag{9}$$

Note that the batch normalization layers use running mean and variance of current batch in training phase while moving averages in testing phase. Although precise $E[y_l']$ and $Var[y_l']$ can be computed in training phase, the moving averages vary dramatically since $b$ in Eq. 9 grows from a small value to a large one during optimization, as shown in Figure 1 (c) where the distribution of $W_l$ with initial variance of $1.0$ is optimized to the one with variance of $88.47$. Thus, the large variation of batch statistics results in the instability of testing loss curve while the training loss curve is similar to the one of CNNs, as shown in Figure 1 (a).

## 3.1 Adaptive Weight Normalization for AdderNet

Based on these observations and analysis, we propose to make some efficient modifications to current ANN optimization. From Eq. 9, the running mean and variance mainly depend on the standard deviation of adder weights so that large magnitude of weights could destabilize the statistics in batch normalization, which indicates that adder weights need normalization to prevent them from being updated to a distribution with large variance. A naive approach is Weight Standardization proposed by Qiao *et al.*[18] as

$$W_{i,j}' = \frac{W_{i,j} - \mu_{W_i}}{\sigma_{W_i}}, \text{ where } \mu_{W_i} = \frac{1}{n} \sum_{j=1}^{n} W_{i,j},$$
$$\sigma_{W_i} = \sqrt{\frac{1}{n} \sum_{j=1}^{n} (W_{i,j} - \mu_{W_i})^2 + \epsilon}, \tag{10}$$

where $W \in \mathbb{R}^{c_{out} \times n}$ denotes the permuted adder weight, $\|$ denotes concatenation operation and $\epsilon$ is added for numerical stability. In Eq. 10, each output channel of adder weight is normalized to a distribution with a mean of $0$ and variance of $1$, which stabilizes the running mean and variance of batch normalization layer according to Eq. 9. However, with weight standardization directly applied to ANNs, there exists a dramatic accuracy drop. For example, ANN with weight standardization achieves $90.62\%$ in ResNet-20 on CIFAR-10 while original ANN achieves $91.84\%$, which causes $1.22\%$ accuracy drop. Although adder weights are normalized to guarantee a stable test phase, rigorous mean and variance are strictly assigned to adder weights, which constrains the representation power of ANNs. Since the similarity between filter and input feature is measured by $\ell_1$-distance in ANNs, the magnitude of weight values can be rather sensitive for network expression capability. Thus, directly applying weight standardization to ANNs can be easily stunk in the local optimum without exploring wider space of adder weights. Instead, we propose to normalize adder weights with trainable variables for each output channel, which preserves the representation power. Eq. 10 can be rewritten as

$$W'_{i,j} = \nu_i \frac{W_{i,j} - \mu_{W_i}}{\sigma_{W_i}} + \upsilon_i, \tag{11}$$

where $\nu_i$ and $\upsilon_i$ are trainable variables similar to $\beta$ and $\gamma$ in batch normalization layer. Thus, the magnitude of weight values can be automatically adjusted to fit the potential levels of freedom of adder weights. Furthermore, previous analysis demonstrates that the magnitude of gradient w.r.t the input $X$ and the filter $W$ in ANNs is much smaller than that in CNNs [1]. With the incorporation of these two parameters, the gradient w.r.t the filter $W$ can be automatically adjusted. The gradient of loss $\ell$ w.r.t the weight $W_{i,j}$ is computed as

$$\frac{\partial \ell}{\partial W_{i,j}} = \sum_{c=1}^{n} \frac{\nu_i}{n^2 \sigma_{W_i}} \left\{ \frac{\partial \ell}{\partial W'_{i,j}} - \frac{\partial \ell}{\partial W'_{i,c}} [1 + \frac{(W_{i,j} - W_{i,c})(W_{i,c} - \mu_{W_i})}{\sigma_{W_i}}] \right\}. \tag{12}$$

In Eq. 12, the gradient of $W$ is amplified by $\nu_i$, which relieve the gradient reduction in ANNs.

## 4 Activate Potential Adversarial Robustness

Consider the intermediate layer forwarding of ANNs and CNNs with perturbation $\delta_l$, the disturbance of an element on the output feature map before BN layer can be computed as

$$\tilde{y}'_{lCNN} - y'_{lCNN} = \sum_{i=1}^{n} [(x_{l,i} - \delta_{l,i}) \times W_{l,i} - x_{l,i} \times W_{l,i}] = \sum_{i=1}^{n} (-\delta_{l,i}) \times W_{l,i},$$

$$\tilde{y}'_{lANN} - y'_{lANN} = \sum_{i=1}^{n} [|x_{l,i} - W_{l,i}| - |x_{l,i} - \delta_{l,i} - W_{l,i}|] \approx \sum_{i=1}^{n} \pm |\delta_{l,i}|, \tag{13}$$

where $\pm$ denotes the choices of two possible signs including addition and subtraction. Note that $|x_{l,i} - W_{l,i}|$ follows a distribution with large mean and variance in ANNs. We assume that $\delta_l$ follows a Gaussian distribution with zero mean and smaller variance $\mathcal{N}(0, \sigma_\delta^2)$ where $\sigma_\delta^2 < Var[W_l]$, with which the subtraction $|x_{l,i} - W_{l,i}| - |x_{l,i} - \delta_{l,i} - W_{l,i}|$ has a high probability to be either $\delta_l$ or $-\delta_l$. We show that it is difficult for $\pm |\delta_{l,i}|$ to have large variance in ANNs under adversarial attacks. In Eq. 13, if $Var[\pm |\delta_{l,i}|]$ grows in ANNs, all the elements in $\pm |\delta_{l,i}|$ tend to select different signs, which automatically eliminate each other to make $\tilde{y}'_{lANN} - y'_{lANN} \approx 0$. Thus, if attacks succeed, $Var[\pm |\delta_{l,i}|]$ needs reduction to guarantee larger perturbation on feature map. To ensure maximum disturbance, all the elements in $\pm |\delta_{l,i}|$ have the same signs, which forms a sum of $n$ half-normal distributions. On the contrary, each $\delta_{l,i}$ in CNNs is transformed by different $W_{l,i}$ in Eq. 13. The variance of disturbance before BN layer can be computed as

$$Var[\tilde{y}'_l - y'_l]_{CNN} = Var[\sum_{i=1}^{n} (-\delta_{l,i}) \times W_{l,i}] = n\sigma_\delta^2 (Var[W_l] + (E[W_l])^2),$$

$$Var[\tilde{y}'_l - y'_l]_{ANN} \approx Var[\sum_{i=1}^{n} |\delta_{l,i}|] = n\sigma_\delta^2 (1 - \frac{2}{\pi}). \tag{14}$$

Eq. 14 indicates the output disturbance variation in CNNs depends on both the statistics of $W$ and $\delta$, and could vary for different output channels while the one in ANNs only depends on the statistics of $\delta$ for all the output channels. The major difference lies in BN layer where both ANN and CNN disturbances are rescaled and the variance of perturbation on next layer can be computed as

$$Var[\delta_{l+1}]_{CNN} = n\sigma_\delta^2 (Var[W_l] + (E[W_l])^2) / [Std(\tilde{y}'_{lCNN})]^2,$$

$$Var[\delta_{l+1}]_{ANN} \approx n\sigma_\delta^2 (1 - \frac{2}{\pi}) / [Std(\tilde{y}'_{lANN})]^2. \tag{15}$$

Note that the variance of $\tilde{y}'_{lANN}$ is much larger than $\tilde{y}'_{lCNN}$. Thus, ANNs have a much smaller disturbance variance on $l + 1$ layer, which suggests that the perturbations of all the elements on $\tilde{y}'_l$ are similar and can be eliminated through subtraction of a scalar value while CNNs cannot copy that success. To activate the adversarial robustness through utilizing this property of ANNs, a simple yet effective method is utilizing the running mean in batch normalization layer for automatic disturbance

Table 1: Adversarial robustness on CIFAR-10 under white-box attacks without adversarial training. -R denotes robust inference strategy which uses the running mean in batch normalization layer instead of tracked ones. BIM[7] denotes iterative attack with 7 steps. The best results in bold and the second best with underline.

| Model | Method | #Mul. | #Add. | Clean | FGSM | BIM[7] | PGD[7] | MIM[5] | RFGSM[5] |
|---|---|---|---|---|---|---|---|---|---|
| ResNet-20 | CNN | 41.17M | 41.17M | **92.68** | 16.33 | 0.00 | 0.00 | 0.01 | 0.00 |
| | ANN | 0.45M | 81.89M | 91.72 | 18.42 | 0.00 | 0.00 | 0.04 | 0.00 |
| | CNN-R | 41.17M | 41.17M | 90.62 | 17.23 | 3.46 | 3.67 | 4.23 | 0.06 |
| | ANN-R | 0.45M | 81.89M | 90.95 | 29.93 | 29.30 | 29.72 | 32.25 | 3.38 |
| | ANN-R-AWN | 0.45M | 81.89M | 90.55 | **45.93** | **42.62** | **43.39** | **46.52** | **18.36** |
| ResNet-32 | CNN | 69.12M | 69.12M | **92.78** | 23.55 | 0.00 | 0.01 | 0.10 | 0.00 |
| | ANN | 0.45M | 137.79M | 92.48 | 35.85 | 0.03 | 0.11 | 1.04 | 0.02 |
| | CNN-R | 69.12M | 69.12M | 91.32 | 20.41 | 5.15 | 5.27 | 6.09 | 0.07 |
| | ANN-R | 0.45M | 137.79M | 91.68 | 19.74 | 15.96 | 16.08 | 17.48 | 0.07 |
| | ANN-R-AWN | 0.45M | 137.79M | 91.25 | **61.30** | **59.41** | **59.74** | **61.54** | **39.79** |

elimination. In Eq. 9, $\tau_\mu$ becomes $\tau_\mu - E[\delta]$ while $b + \tau_\sigma$ nearly remains the same under attack settings since the disturbance has small variance. The forwarding of BN layer can be computed as

$$
\begin{aligned}
\tilde{y} &= \gamma \frac{\tilde{y}' + n(b + \tau_\sigma + |\tau_\mu - E[\delta]|)}{\sqrt{n[(b+\tau_\sigma)^2 - 2|\tau_\mu - E[\delta]|(b+\tau_\sigma)]}} + \beta \\
&\approx \gamma \frac{y' + \sum_i^n \pm|\delta_i| + n(b + \tau_\sigma + |\tau_\mu|) - n|E[\delta]|}{\sqrt{n[(b+\tau_\sigma)^2 - 2|\tau_\mu - E[\delta]|(b+\tau_\sigma)]}} + \beta \\
&\approx \gamma \frac{y' + n(b + \tau_\sigma + |\tau_\mu|)}{\sqrt{n[(b+\tau_\sigma)^2 - 2|\tau_\mu - E[\delta]|(b+\tau_\sigma)]}} + \beta.
\end{aligned}
\tag{16}
$$

The difference between Eq. 9 and Eq. 16 lies in the running variance, which demonstrates that the disturbance can be significantly relieved through computing the running mean in BN layer while the variance is expected to be constant to eliminate the difference brought by $E[\delta]$. Thus, we propose ANN robust inference strategy as

$$
y^* = \gamma \frac{\tilde{y} - \text{running mean}}{\text{tracked variance}} + \beta.
\tag{17}
$$

However, considering the actual case where the signs of $\pm|\delta_{l,i}|$ are not the same everywhere, the proposed ANN robust inference strategy cannot work appropriately and results in residual perturbations which will be rescaled by tracked variance after BN layer. In Eq. 13, the perturbations of different output channels are similar in ANNs, which indicates that the noise of feature map can hardly cross channels. However, this property will be broken if tracked variances have enormous differences among channels. Thus, our proposed ANN-AWN can successfully relieve the variation across channels to further improve adversarial robustness, which is verified in later empirical evaluations.

# 5 Experiments

In this section, we empirically evaluate the superiority of the proposed approach on different tasks and datasets, including the adversarial robustness, object detection and optimization stability.

## 5.1 Adversarial Robustness Evaluation

**White-box Attacks Setup.** To demonstrate how the proposed adaptive weight normalization activates the potential adversarial robustness of AdderNet, we conduct a series of experiments. Following [1], we first train both CNNs and ANNs with ResNet20 and ResNet32 on CIFAR-10 under the same settings. CIFAR-10 dataset contains $50K$ training images and $10K$ validation images with size of $32 \times 32$ over 10 classes. We use SGD optimizer with an initial learning rate of 0.1, momentum of 0.9 and a weight decay of $5 \times 10^{-4}$. The model is trained on single V100, which takes 400 epochs with a batch size of 256 and a cosine learning rate schedule. The learning rate of trainable parameter $\nu$ and $\upsilon$ in AWN is rescaled by a hyper-parameter which we set to be $1 \times 10^{-5}$. For adversarial robustness evaluation, we conduct white-box attacks on these models

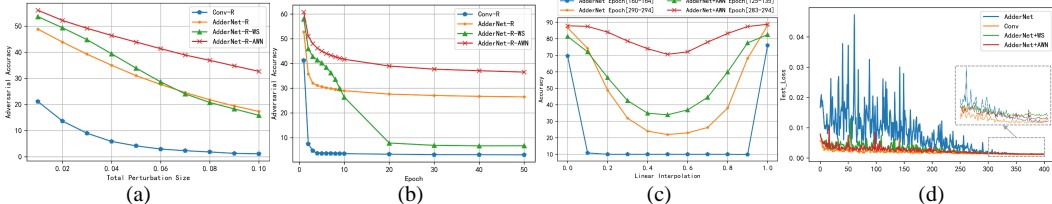

| (a) | (b) | (c) | (d) |

Figure 2: The evaluation of adversarial robustness under different PGD attack size is shown in (a) and different PGD attack steps in (b). The performance of intermediate weights sampled from ANN and ANN-AWN through linear interpolation in (c). Test loss curves of CNN, ANN, ANN-WS and ANN-AWN are shown in (d).

including Fast Gradient Sign Method (FGSM) [24], Basic Iterative Method (BIM) [14], Projected Gradient Descent (PGD) [17], Momentum Iterative Method (MIM) [6], and RFGSM [27] to generate adversarial examples. Following previous adversarial literature [17, 31], the adversarial perturbation is considered under $l_\infty$ norm with the total perturbation size of $8/255$. In iterative attack settings, the step size is set to $2/255$. The number of iterations is set to 7 for PGD and BIM attacks, and 5 for MIM and RFGSM attacks. Note that different from traditional defense algorithms which generate adversarial samples for adversarial training [21] or search robust architectures [5, 15], our proposed algorithm utilizes the properties of ANNs to achieve adversarial robustness without any training tricks or modifications of architecture. The experimental results are shown in Table 1. Note that all the models are trained with clean images, which dismisses the expensive adversarial training.

**Against White-box Attacks.** As shown in the first two rows in Table 1, these adversarial attacks successfully mislead both CNN and ANN to provide wrong predictions. Although ANN has slightly better adversarial accuracy compared to CNN, such as $0.01\% \rightarrow 0.04\%$ under MIM[5] attack, the potential adversarial robustness shown in Eq. 13 cannot be activated with normal settings. Thus, we replace the tracked mean in batch normalization layer with the running mean of current batch based on Eq. 17 and compare ANN with CNN in the third and fourth rows, which are denoted as -R. Comparing CNN-R and ANN-R, there exists an enormous adversarial robustness improvement. For example, ANN-R achieves **26.05**% accuracy increment from $3.67\% \rightarrow 29.72\%$ under PGD[7] compared with CNN-R, and improves the accuracy by **32.21**% from $0.04\% \rightarrow 32.25\%$ under MIM[5] compared with ANN. The comparison between ANN and ANN-R empirically demonstrates that appropriate utilization of the running mean in batch normalization layer of ANNs can significantly activate the adversarial robustness of ANNs. Furthermore, the comparison between CNN-R and ANN-R shows strong evidence of the natural robustness difference between CNNs and ANNs and indicates that the $\ell_1$ distance and the independence between perturbation and adder weight provides much better defense than CNNs, which is consistent with the aforementioned variance analysis. The evaluation of proposed AWN is shown in the fifth row. For all the attacks, ANN-R-AWN achieves the best results, which demonstrates the effectiveness of proposed AWN. Comparing ANN-R and ANN-R-AWN, AWN shows obvious superior adversarial robustness. For example, AWN improves adversarial accuracy by **13.67**% from $29.72\% \rightarrow 43.39\%$ under PGD[7] and **14.98**% from $3.38\% \rightarrow 18.36\%$ under RFGSM[5], which illustrates that adversarial robustness can be further boosted through narrowing the difference among channels to relieve the perturbation transformation. On ResNet-32, ANN-R achieves worse performance than the one with ResNet-18, which demonstrates that proposed robust inference strategy is not sufficient for superior adversarial robustness. However, ANN-R-AWN consistently achieves better performance, which outperforms other baselines in all the adversarial scenarios, which indicates that the robustness of proposed AWN can generalize to deeper models.

**Against Gradually Enhanced Attacks.** We highlight the superiority of proposed ANN-R-AWN through enhancing the attacks from different aspects to evaluate the adversarial robustness under more powerful attacks. We use the naturally trained models with ResNet-20 on CIFAR-10 for evaluation. In the first scenario, the total perturbation size $\epsilon$ of PGD attack increases from 0.01 to 0.1 with a step size of $\epsilon/7$. In the second scenario, the iterations of PGD attacks are enhanced from 1 to 50. The evaluation results are shown in Figure 2 (a) and (b). ANN-R-AWN obtains much better defense ability than ANN-R, CNN-R and ANN-R-WS baselines when the attack size grows. For example, the adversarial accuracy of CNN-R quickly dive to $2.96\%$, ANN-R to $27.67\%$ and ANN-R-WS to $28.66$ when the PGD attack size reaches $0.06$ while ANN-R-AWN maintains $32.67\%$ when the PGD attack size reaches $0.1$. Similarly, ANN-R-AWN also achieves superior robustness over other baselines with increasing steps of PGD attack. For example, the adversarial accuracy of ANN-R

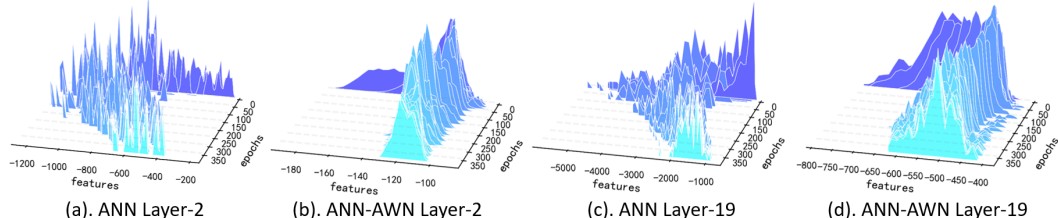

| (a). ANN Layer-2 | (b). ANN-AWN Layer-2 | (c). ANN Layer-19 | (d). ANN-AWN Layer-19 |

Figure 3: Distributions of features from different layers of ANN and ANN-AWN on different epochs.

under PGD$^{50}$ becomes $26.48\%$ while ANN-R-AWN maintains $36.52\%$ with $\mathbf{10.04\%}$ increment, which demonstrates the effectiveness of AWN. The enormous accuracy drop of ANN-R-WS along with attack epoch and coherent robustness of our proposed AWN illustrate the necessity of introduced adaptive rescale and shift parameters in weight normalization. Thus, the advantage of AWN becomes more obvious under more powerful attacks, which highlights the superiority of proposed AWN.

## 5.2 Stability of AdderNets

**Stability Evaluation through Linear Interpolation.** To conduct a quantitative evaluation of AdderNet stability, we propose to evaluate the smoothness of optimization landscape through tracking the performance of AdderNet parameters which are sampled between different training epochs. To highlight the stability of our proposed AWN, we select adjacent epochs for ANN and ANN-AWN trained with ResNet-20 on CIFAR-10 and sample 9 intermediate weights between two epochs through linear interpolation. The comparison is shown in Figure 2 (c). To evaluate the stability of early training stage, we select epoch $160, 164$ for ANN as the blue curve and select epoch $125, 135$ for ANN-AWN as the green curve. Although we set earlier epoch with larger intervals for ANN-AWN, our proposed algorithm forms a much more smooth curve with a smaller variance $336.00$ compared to the one of ANN $587.28$. In the later training stage, we select epoch $290, 294$ for ANN as the orange curve and select epoch $283 - 294$ for ANN-AWN as the red curve. Comparing with ANN, our proposed algorithm achieves variance reduction of $585.97$ from $627.54$ to $41.67$. The performance of parameters sampled from linear interpolation reflects the stability of models and the smoothness of the optimization landscape. With the proposed AWN, the optimization of AdderNet can be significantly stabilized, which could benefit the situations where adder layers are difficult to optimize.

**Trade-offs Between Stability and Accuracy.** To further illustrate how AWN takes effect, we keep track of the test loss during AdderNet optimization and the results of both CNN and ANN with ResNet-20 on CIFAR-10 are shown in Figure 2 (d). The test loss curve of AdderNet is quite unstable before 300 epochs while CNNs, ANN with weight standardization and ANN with adaptive weight normalization all achieve relatively smoothed loss curve, which indicates that the instability of AdderNet mainly comes from the large variance of adder weights and the proposed AWN eliminate it successfully. The dotted area covers from 300 to 400 epochs. Although AdderNet cannot achieve stability due to the large variance of weights, ANNs can still achieve similar performance as CNNs since the variation of adder weights is reduced during the end of training. However, WS fails to achieve a similar classification performance and is stuck at local optimum, which demonstrates that the normalization on adder weights could hurt the expressive power. On the contrary, our proposed AWN achieves relatively better performance through incorporating the adaptive trainable parameters for adder weights, which enables them to shift and rescale back to restore the original performance.

**Feature Distribution Analysis.** We visualize and track the feature distributions of AdderNet with or without AWN during the training, as shown in Figure 3. We randomly sample features before batch normalization layer at different epochs and compute histograms over them. Both ANN and ANN-AWN feature distributions in the 2nd and 19th layers are shown in Figure 3 (a),(b),(c) and (d) respectively. ANN feature distributions vary dramatically during the optimization, which enormously disturbs the tracking of mean and variance in batch normalization layer. For example, the sampled feature on 21-th epoch in (c) has mean of $-2738.58$ and standard deviation of $1042.44$ while the one on 386-th epoch has mean of $-1476.52$ and standard deviation of $266.97$ with $1262.06$ increment on mean and $775.47$ reduction on standard deviation. However, those in ANN-AWN become much milder, which stabilizes ANNs. For example, the sampled feature on 21-th epoch in (d) has mean of $-940.89$ and standard deviation of $269.39$ and the one on 385-th epoch has mean of $-564.59$ and standard deviation of $60.65$ with $376.3$ increment on mean and $208.74$ reduction on standard

Table 2: ANN stability evaluation on object detection task. Comparison of proposed approach on ANNs with other settings on PASCAL VOC 2012 benchmark. The $[\cdot]$ in backbone denotes the classification accuracy of pretrained network on ImageNet.

| Model | Backbone | Neck | mAP |
|---|---|---|---|
| CNN-FPN | Res18-CNN [69.8] | CNN | 69.3 |
| ANN-FPN | Res18-ANN [67.0] | ANN | 68.6 |
| ANN-WS-FPN | Res18-ANN-WS [64.1] | ANN-WS | 67.0 |
| ANN-AWN-FPN | Res18-ANN-AWN [67.1] | ANN-AWN | 69.4 |

Table 3: Adversarial robustness comparison of WS and AWN under different inference strategy with ResNet-20 on CIFAR-10 with natural training. -R denotes using running mean of current batch in BN layer. -r denotes using both running mean and variance.

| Model | Clean | FGSM | BIM[7] | PGD[7] | MIM[5] | RFGSM[5] |
|---|---|---|---|---|---|---|
| ANN-r-WS | 88.10 | 49.24 | 21.46 | 22.96 | 32.26 | 7.31 |
| ANN-r-AWN | 89.81 | **49.85** | 22.54 | 24.27 | 32.54 | 8.26 |
| ANN-R-WS | 89.38 | 40.65 | 31.35 | 36.08 | 42.3 | 7.78 |
| ANN-R-AWN | **90.55** | 45.93 | **42.62** | **43.39** | **46.52** | **18.36** |

deviation. Furthermore, the difference among channels are effectively reduced by AWN to constrain the perturbation space of adversarial examples, *e.g.*, the standard deviation of ANN features on the second layer after training becomes $87.41$ while that of ANN-AWN becomes $31.13$ with a massive reduction, which potentially provides better defense ability against more powerful adversarial attacks.

## 5.3 Experiments on Object Detection

To further illustrate the advantage of imposing stability on AdderNets, we conduct experiments on object detection with ANNs on PASCAL VOC (VOC) dataset. VOC contains 20 object classes, the training set includes 10K images which are the union of VOC 2007 and VOC 2012, and the VOC 2007 test set with 4.9K images is used for evaluation. The mAP scores using Iou at 0.5 are reported. All the models are trained with the same setting. Based on the variance study of ANNs, we unfreeze BatchNorm during the training. Following [2], we insert more shortcuts in the neck part. We use an initial learning rate of $0.008$ with a linear warmup for $500$ iterations, momentum of $0.9$, weight decay of $1 \times 10^{-4}$ and a cosine learning rate strategy. All the models are trained on 4 V100 GPUs with SGD optimizer for 12 epochs with a batch size of 4. For the detector baseline, we include both CNN and vanilla ANN for comparison. ANN-FPN replaces the convolution layers with adder layers in the pretrained ResNet-18 backbone and neck of Faster R-CNN [20]. Through applying different types of ANNs to detection, we conduct comparison among the CNN, vanilla ANN [1], ANN-WS and ANN-AWN. The detailed evaluation is shown in Table 2. In Backbone column, the number in brackets denotes the classification accuracy pretrained on ImageNet. Comparing ANN with our proposed AWN, although they achieve similar classification performance, AWN improves the mAP score by $\mathbf{0.8}$ from $68.6 \rightarrow 69.4$. Even comparing with CNN-FPN which has superior classification performance, our proposed AWN still outperforms it, which demonstrates the necessity of stability. With a much more smooth loss landscape, the optimization of AdderNet on other tasks can be easier and more stable. However, ANN-WS is not competitive with other baselines, with $\mathbf{2.4}$ mAP reduction compared with AWN, which empirically verifies that directly normalizing adder weights could limit the ability of feature extraction and performance for other tasks. Note that there exists an enormous accuracy drop of pretrained ANN-WS, which significantly constrains its detection performance. Thus, besides stability, the representation power of ANNs can be rather important in terms of applying ANNs to other tasks. On the contrary, our proposed AWN achieves better trade-offs between classification performance and stability through an adaptive scheme, which together achieves the superior mAP score in detection task.

## 5.4 Ablation Studies

We conduct ablation studies on proposed ANN-R-AWN to illustrate the effectiveness of adaptive training parameters and proposed robust inference strategy. We have already shown that weight standardization can easily be stuck at a local optimum in Figure 2 (d). Although the analysis in Sec 4 works for both WS and AWN, we empirically verifies that the gaps of clean accuracy between WS and AWN still exist in adversarial accuracy. The results are shown in Table 3 where WS and AWN are further attacked and evaluated to demonstrate the influence of performance drop on clean and adversarial accuracy. Comparing WS and AWN, our proposed AWN consistently outperforms WS in both adversarial and clean accuracy, which indicates that directly normalizing adder weights could hurt the representation power of ANNs and restrict the adversarial accuracy of ANNs. With proposed parameters $\nu$ and $\upsilon$ in Eq. 11, AWN successfully relieves this problem through exploring the balance between expressive power and weight magnitude reduction, which achieves better classification

performance and adversarial robustness. We further evaluate the effectiveness of proposed ANN robust inference strategy as in Eq. 17. We denote the strategy which replaces both tracked mean and variance with running ones as -r and our proposed one as -R. Comparing two inference methods, our proposed strategy consistently outperforms -r, which verifies that the activated robustness mainly comes from the running mean which automatically eliminates the perturbations.

## 6  Conclusion

In this paper, we investigate the major concerns of AdderNets through approximating the mean and variance of output features of an arbitrary adder layer. With a derived lower boundary, we show that the instability of AdderNets mainly comes from drastic fluctuations of running mean and variance in batch normalization layer whose computation is dominated by the variance of weights. Our proposed adaptive weight normalization (AWN) works with AdderNets to optimize adder weight distributions adaptively, which significantly improves the stability and leads to smooth landscape. Our analysis of the adder layer forwarding reveals the potential superior defense ability of AdderNets against perturbations and proposed robust inference strategy together with AWN successfully activate the adversarial robustness. Experiments conducted on stability and robustness demonstrate the superior performance of the proposed ANN-AWN.

## Acknowledgment

The authors would like to thank the Area Chair and the reviewers for their constructive comments. This work was supported in part by the Australian Research Council under Projects DE180101438 and DP210101859.

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
