# Towards Stable and Robust AdderNets
# (Supplementary Material)

**Minjing Dong**[1,2], **Yunhe Wang**[2],[*] **Xinghao Chen**[2], **Chang Xu**[1]
[1]School of Computer Science, University of Sydney
[2]Huawei Noah's Ark Lab
mdon0736@uni.sydney.edu.au, yunhe.wang@huawei.com,
xinghao.chen@huawei.com, c.xu@sydney.edu.au

## A  Stability of Robustness

Different from traditional defense methods which are mainly based on adversarial training, our proposed robust inference strategy works during testing phase. Since our proposed ANN-R-AWN makes use of the running mean of current batch, the performance could vary due to different inference settings. Here we provide stability test of robustness under different batch size and shuffle settings to eliminate the concerns that the superior adversarial robustness comes from other aspects.

Table 1: Robustness stability of proposed ANN-R-AWN with ResNet-20 on CIFAR-10 under different inference settings.

| Batch size | Shuffle | Clean | FGSM | BIM[7] | PGD[7] | MIM[5] | RFGSM[5] |
|---|---|---|---|---|---|---|---|
| 96 | ✗ | 90.55 | 45.93 | 42.62 | 43.39 | 46.52 | 18.36 |
| 96 | ✓ | 90.59 | 46.81 | 42.26 | 43.42 | 46.65 | 18.19 |
| 64 | ✗ | 90.56 | 45.58 | 41.53 | 42.42 | 46.03 | 18.15 |
| 256 | ✗ | 90.78 | 46.93 | 42.62 | 43.28 | 46.51 | 20.31 |

As shown in Table 1, the stability of adversarial robustness is evaluated under different inference settings. We first show whether the shuffle of test set influences the performance. Comparing the first two rows, ANN-R-AWN achieves similar performance with or without shuffle. Furthermore, we forward test set with different batch size as shown in the last two rows. It is obvious that both clean and adversarial accuracy has slight increment with increasing batch size, which matches the well-known observation that networks with batch normalization layer gets better performance with larger batch size. For adversarial robustness, ANN-R-AWN still outperforms CNN and ANN with a large margin when a smaller batch size of 64 is selected, which illustrates that the superior adversarial robustness mainly comes from proposed ANN robust inference strategy and adaptive weight normalization. For a fair comparison, all the empirical experiments of adversarial robustness are conducted with the same inference setting of a batch size of 96 without shuffle.

## B  Robustness Comparison with Adversarial-trained CNN

In the main body, we mainly focus on the comparison with CNN since we want to highlight the natural robustness of AdderNet compared to CNN under the same setting. We further provide the comparison with other advanced defense techniques on CNN, as shown in Table 2. Adversarial training is one of the most effective approaches for defending adversarial examples and different variants have been proposed, such as PGD-AT [2], ALP [1] and TLA [3]. We evaluate these algorithms under

---

[*]Corresponding author.

35th Conference on Neural Information Processing Systems (NeurIPS 2021).

various attacks with the same settings, such as FGSM, BIM, PGD, MIM as well as CW. Our proposed ANN-AWN-R achieves the best performance since the perturbation brought by adversarial examples can be automatically eliminated by BN layers and attacking space across channels is reduced by AWN (Section 3.3). Note that our algorithm only needs a natural training without feeding any adversarial examples, however, the adversarial robustness of ANN-AWN-R still outperforms CNN with defense techniques, which demonstrates the effectiveness of proposed algorithm on ANNs.

Table 2: Robustness Comparison with CNN defense techniques. AT denotes the usage of adversarial training.

| Method | AT | Clean | FGSM | BIM$^7$ | PGD$^7$ | MIM$^{40}$ | CW$^{30}$ |
|---|---|---|---|---|---|---|---|
| PGD-AT | ✓ | 87.14 | 55.63 | 48.29 | 49.79 | 45.16 | 46.97 |
| ALP | ✓ | 89.79 | 60.29 | 50.62 | 51.89 | 45.97 | 47.69 |
| TLA | ✓ | 86.21 | 58.88 | 52.60 | 53.87 | 50.09 | **50.69** |
| ANN-AWN-R | ✗ | **91.25** | **61.30** | **59.41** | **59.74** | **66.43** | 50.60 |

## C   Adaptive Weight Normalization on CNN

Our proposed Adaptive Weight Normalization is based on the analysis of the variance of ANN features and specifically designed for ANNs. We further evaluate the performance of AWN on CNNs.

Table 3: Adversarial robustness evaluation of AWN on CNN and ANN with ResNet-32 on CIFAR-10.

| Model | Clean | FGSM | BIM$^7$ | PGD$^7$ | MIM$^5$ | RFGSM$^5$ |
|---|---|---|---|---|---|---|
| CNN-R | 91.32 | 20.41 | 5.15 | 5.27 | 6.09 | 0.07 |
| ANN-R | 91.68 | 19.74 | 15.98 | 16.08 | 17.48 | 0.07 |
| CNN-R-AWN | 92.33 | 21.71 | 5.74 | 5.94 | 7.16 | 0.05 |
| ANN-R-AWN | 91.25 | **61.30** | **59.41** | **59.74** | **61.54** | **39.79** |

As shown in Table 3, with the involvement of AWN, CNN obtains slight better adversarial robustness. However, comparing with ANN, the improvement robustness of CNN with AWN is marginal, which demonstrates that the superior collaboration of proposed AWN with ANNs and shows strong evidence of potential robustness of ANNs.

## D   Variants of Inference Strategy

Our proposed ANN robust inference strategy is derived from the analysis of variance in ANNs. To illustrate the effectiveness of proposed inference strategy, we evaluate several variants of them with ResNet-32 on CIFAR-10.

Table 4: Adversarial robustness evaluation of different inference strategy of ANN-AWN with ResNet-32 on CIFAR-10.

| Running Mean | Running Variance | Clean | FGSM | BIM$^7$ | PGD$^7$ | MIM$^5$ | RFGSM$^5$ |
|---|---|---|---|---|---|---|---|
| ✗ | ✗ | 92.26 | 29.70 | 0.05 | 0.08 | 0.84 | 0.01 |
| ✗ | ✓ | 88.95 | 17.23 | 10.49 | 11.69 | 16.74 | 8.63 |
| ✓ | ✗ | 91.25 | **61.30** | **59.41** | **59.74** | **61.54** | **39.79** |
| ✓ | ✓ | 90.38 | 54.69 | 25.65 | 26.55 | 38.56 | 14.03 |

Our proposed robust inference strategy makes use of running mean of current batch and the tracked variance in batch normalization layer, which denotes the third row in Table 4. Comparing with other variants of inference strategy, our proposed one achieves the best robustness with a large margin. The evaluation shows strong evidence that the perturbations can be eliminated by the subtraction of a single scalar value on feature map, which is consistent with our analysis in Section 3.3.

# E  Classification Performance

To illustrate the effectiveness of adaptive weight normalization, we evaluate the classification performance of ANN, ANN-WS and ANN-AWN on CIFAR-10 and ImageNet. Note that our main contribution lies on the stability and robustness of AdderNets. The evaluation of classification are included here for the completeness.

Table 5: Classification performance evaluation.

| Method | Model | Dataset | Accuracy |
|---|---|---|---|
| ANN | ResNet-20 | CIFAR-10 | 91.84 |
| ANN-WS | ResNet-20 | CIFAR-10 | 90.62 |
| ANN-AWN | ResNet-20 | CIFAR-10 | 91.42 |
| ANN | ResNet-18 | ImageNet | 67.00 |
| ANN-WS | ResNet-18 | ImageNet | 64.17 |
| ANN-AWN | ResNet-18 | ImageNet | 67.11 |

The comparison is shown in table 5. The exists a trade-offs between ANN classification performance and stability, as discussed in Section 4.2. As shown in the first three rows, both ANN-WS and ANN-AWN have accuracy drop, however, ANN-WS has relatively larger drop from $91.84\% \rightarrow 90.62\%$ and ANN-AWN has acceptable drop from $91.84\% \rightarrow 91.42\%$. The gap becomes more obvious when methods are evaluated on ImageNet. Our proposed ANN-AWN achieves slightly better performance than ANN while ANN-WS has a dramatic accuracy drop, which empirically verified our analysis in Sec 3.2 that the performance could be largely constrained without incorporating the shift and scale parameters.

# F  Normalization Variants

Besides batch normalization layers used in vanilla ANNs, there exist other types of normalization which dismiss the usage of batch statistics, such as Layer Norm and Group Norm. To demonstrate the necessity of batch normalization, we include other normalization for comparison. We conduct experiments with ResNet-20 on CIFAR-10, as shown in Table 6. After replacing BN with other types including LN and GN, there exist a tremendous accuracy drop of ANNs. Thus, although other types of normalization tackle the problem of instability, the worse performance prevents them from actual usage.

Table 6: Classification performance of normalization variants.

| Normalization Type | Accuracy |
|---|---|
| Batch Norm | 91.84 |
| Layer Norm | 69.72 |
| Group Norm | 80.69 |

# G  Hyper-parameters

The learning rate of trainable parameters in AWN is rescaled by a hyper-parameter. In our empirical evaluations, AWN is quite sensitive to this hyper-parameter since it directly controls the optimization of affine transformation in Eq. 11. A relatively large value of training rate could lead to explosion during training. This hyper-parameter is determined by several trials. An appropriate value of it should be around 1e-5. We provide ablation study of this hyper parameter on CIFAR-10 with ResNet-20, as shown in Table 7.

# References

[1] H. Kannan, A. Kurakin, and I. Goodfellow. Adversarial logit pairing. *arXiv*, 2018.

Table 7: Hyper-parameter study of AWN-ANN.

| LR Rescale Rate | Accuracy |
|---|---|
| 1.0 | NaN |
| 1e-3 | 91.27 |
| 1e-5 | 91.42 |
| 1e-7 | 91.05 |

[2] A. Madry, A. Makelov, L. Schmidt, D. Tsipras, and A. Vladu. Towards deep learning models resistant to adversarial attacks. In *ICLR*, 2018.

[3] C. Mao, Z. Zhong, J. Yang, C. Vondrick, and B. Ray. Metric learning for adversarial robustness. In *NeurIPS*, pages 480–491, 2019.