# OpenReview forum: "Towards Stable and Robust AdderNets"
_NeurIPS.cc/2021/Conference — NeurIPS 2021 Poster_

### Official Review · Reviewer_oKa2 · 2021-07-13

**Rating:** 7
**Confidence:** 4

**Summary:**

This paper investigates the mean and variance of features for AdderNets and reveals that the instability of AdderNets mainly comes from fluctuations of running mean and variance in batch normalization layers. To this end, this paper proposes the adaptive weight normalization (AWN) to optimize adder weight distributions adaptively. Extensive experiments demonstrate that the proposed method improves AdderNets on both stability and robustness.

**Limitations And Societal Impact:**

Please discuss a bit how AdderNets affect the future design of neural networks and corresponding chips since they are tailored for energy-efficient networks.

**Main Review:**

Pros:
1.	This paper is technically sound and well written.
2.	The motivation of the proposed method is clear and the proposed idea is quite interesting. The theatrical analysis well justifies the proposed adaptive weight normalization.
3.	The analysis of adversarial robustness for AdderNets makes sense and the proposed method for improving robustness is simple yet effective.
4.	The proposed method shows strong performance for adversarial robustness and presents good stability for downstream tasks like object detection.

Cons:

1.	In Line 193, the learning rate of trainable parameters in AWN is rescaled by a hyper-parameter 1e-5. How is this parameter determined? Are there more ablation studies for the impact of this hyper-parameter?
2.	The concatenation operations in Eq. (10) and Eq. (11) seem a bit weird and I think the concatenation is not necessary. Please carefully check the formulations.
3.	Line 10: “weigh normalization” should be “weight normalization”. Please carefully proofread the manuscript.

**Time Spent Reviewing:**

36

---

> ### Author Response · Authors · 2021-08-10
> **Response to Reviewer oKa2**
>
> Thanks for your constructive comments and support.
>
> **The learning rate of trainable parameters in AWN is rescaled by a hyper-parameter 1e-5. How is this parameter determined? Are there more ablation studies for the impact of this hyper-parameter?**
>
> AWN is quite sensitive to this hyper-parameter since it directly controls the optimization of affine transformation in Eq. 11. A relatively large value of training rate could lead to explosion during training. This hyper-parameter is determined by several trials. An appropriate value of it should be around 1e-5. We provide ablation study of this hyper parameter on CIFAR-10 with ResNet-20:
>
> |LR Rescale Rate| Acc |
> |-----|-----|
> | 1.0 | NaN |
> | 1e-3 | 91.27% |
> | 1e-5 | 91.42% |
> | 1e-7 | 91.05% |
>
>
> **The concatenation operations in Eq. (10) and Eq. (11) seem a bit weird and I think the concatenation is not necessary. Line 10: “weigh normalization” should be “weight normalization”.**
>
> Thanks for pointing it out. We will fix them in the final version.

---

### Official Review · Reviewer_8kiu · 2021-07-16

**Rating:** 6
**Confidence:** 4

**Summary:**

This paper aims at enhancing the stability of AdderNets for performance improvement in downstream tasks. For an adder layer, approximations of feature variance and expectation are derived, with which the variance of weight is found to be the major cause of instability. Adaptive weight normalization (AWN) is thus introduced for addressing this problem. Together with the feature variance approximation, the potential adversarial robustness of AdderNets is also analyzed through comparing the perturbation variance in both AdderNets and CNNs. A simple robust inference strategy with BN layer is thus designed for disturbance elimination.

**Limitations And Societal Impact:**

The limitations of AdderNets have been discussed by the authors in the introduction. For AWN, there exist trade-offs between stability and classification performance, which is discussed in the Sec E of supplementary material.

**Main Review:**

I think this paper provides a good analysis on the variance and exposes an important problem in AdderNets that the instability could constrain the performance of other downstream tasks, such as object detection which requires pre-trained AdderNets. Proposed AWN normalizes weights and involves affine transformation to explore the trade-offs between representation power and stability, which can be treated as a modified version of weight standardization. Extensive experiments on stability evaluation and object detection task demonstrates the necessity of proposed method.

The further discussion of adversarial robustness seems interesting. Due to the difference of addition and multiplication, the paper provides a contrastive analysis of disturbance in AdderNets and CNNs to highlight the potential robustness in AdderNets. The BN statistics in Eq. 9 is further extended to the one under attack in Eq. 16, where the disturbance can be automatically eliminated by the running mean in BN layer, which seems efficient and effective.

Overall, the paper is a good contribution to Adder Neural Networks and adversarial robustness. My major concerns come from the comparison with WS. Proposed AWN is apparently a modified version of existing WS, however, the reason why such a modification is necessary is not fully discussed. Based on the analysis in Eq. 8 and Eq. 9, the instability is highly correlated with the variance of weight, which can be solved through WS. Similarly, WS also holds true for the potential adversarial robustness discussed in Eq. 16. Although I notice that all the empirical evaluations demonstrate that AWN performs better than WS in Table 1-3 and Fig 2, the key factor which brings this improvement is not well-explained in the paper.


**Time Spent Reviewing:**

4 hours.

---

> ### Author Response · Authors · 2021-08-10
> **Response to Reviewer 8kiu**
>
> Thanks for your constructive comments and support.
>
> **My major concerns come from the comparison with WS.**
>
> We agree that WS holds true for the analysis in Eq. 8, 9 and 16. However, there exist an obvious gap on the classification performance between ANN-WS and vanilla ANN due to its limited representation capability. For example, WS reduces the accuracy by 1.22% on CIFAR-10 with ResNet-20 and the gap becomes more critical on ImageNet with 2.83% reduction, as shown in Table 5 in our supplementary material. We attribute this gap to the standard normalization of weight in ANNs. If the weights in ANNs are strictly normalized, the range of output feature values is largely reduced. Note that the basic computation between feature F and filters W in ANNs is addition instead of multiplication, which indicates that the output range mainly depends on the one with larger absolute value between F and W. However, W is strictly normalized in WS, which limits the representation power of ANNs. Thus, we introduce an adaptive weight normalization to tackle this issue through enabling W to have different ranges of values. To demonstrate the necessity of this modification, we include WS as a baseline in all our empirical evaluations, including object detection task in Table 2 and adversarial robustness evaluation in Table 3 and Figure 2, and all the experiments demonstrate the effectiveness of proposed AWN.

---

### Official Review · Reviewer_VLQB · 2021-07-16

**Rating:** 7
**Confidence:** 5

**Summary:**

The paper reveals the existing instability of Adder Neural Networks (ANNs) through the variance estimation of output features and proposes adaptive weight normalization to tackle this issue. Additionally, the adversarial robustness of ANNs is highlighted through further analysis of perturbation variance in both ANNs and CNNs, and the paper proposes to automatically eliminate the perturbations based on the statistics in the batch normalization layer. Experiments on several tasks validate the effectiveness of the proposed method in improving the stability and robustness of ANNs.

**Ethical Concerns:**

No ethical concern

**Limitations And Societal Impact:**

The proposed ANN-R-AWN seems sensitive to the batch size since the inference strategy involves running mean of batch normalization layer, which has been discussed by the authors.

There is no immediate social impact.

**Main Review:**

Pros:
- The entire paper is easy to follow. The observations in ANNs seem interesting and the motivation is straightforward, which aims at eliminating instability of batch normalization layers according to the lower boundary of feature variance in Eq. 8. The variance study in Section 3 seems well-organized and reasonable. The proposed adaptive weight normalization then comes naturally. Through the discussion of perturbation influence in ANNs compared with CNNs, the paper explains that batch normalization layers can form automatic perturbation eliminations and introduced a simple yet effective inference strategy in Eq. 17.
- Sufficient experimental results are provided to verify the effectiveness of proposed adaptive weight normalization for improving ANN stability on object detection tasks and stability studies in Sections 5.2 and 5.3. Adversarial robustness is also evaluated through white-box attacks in Table 1. All the experiments show obvious superiority in both network stability and robustness over other baselines.


Cons:

- My main concern is about the normalization of ANNs. The analysis in Section 3 indicates that the instability mainly comes from the statistics of batch normalization layers in Eq. 9. However, there exist various normalization layers without the statistics involving batch, such as layer normalization. Although the analysis shows that a large variance of feature comes from the weight parameters, I wonder why not directly apply the aforementioned normalization variants to ANNs for improving stability? This discussion or comparison is missing in the paper.

Additionally, some texts in figures are too small to read, such as Figures 1 and 2.


**Time Spent Reviewing:**

3

---

> ### Author Response · Authors · 2021-08-10
> **Response to Reviewer VLQB**
>
> Thanks for your constructive comments and support.
>
> **I wonder why not directly apply the aforementioned normalization variants to ANNs for improving stability? This discussion or comparison is missing in the paper.**
>
> The selection of normalization layer is mainly decided by empirical evaluations. After the variance studies of ANNs, we have tried different types of normalization layers to eliminate the instability of BN layers, including group normalization and layer normalization. We provide the classification performance of ANNs applied with different normalization layers on CIFAR-10 with ResNet-20. The results are reported in the following table:
>
> |BN Type|Acc|
> |-----|-----|
> | Batch Norm | 91.84% |
> | Layer Norm | 69.72% |
> | Group Norm| 80.69% |
>
> Unlike CNNs, ANNs are quite sensitive to the type of normalization layers according to our experiments. The stability at a cost of massive accuracy reduction is inacceptable. Although there exist various normalization layers which dismiss batch statistics to provide stability in ANNs, these variants have relatively large accuracy reductions, which makes them difficult to be deployed to downstream tasks. Thus, instead of directly replacing BN layers with other variants, we focus on alleviating the instability of BN statistics through controlling the variance of weight parameters in ANNs.
>
> **Some texts in figures are too small to read, such as Figures 1 and 2.**
>
> Thanks for your suggestion. We will fix them in the final version.

---

### Official Review · Reviewer_xDpY · 2021-07-19

**Rating:** 6
**Confidence:** 4

**Summary:**

This is an interesting paper to discuss the variance in the adder neural network. The authors have an in-depth comparison between ANN and CNN. Also, the adder operation in the neural network can enjoy some nature advantage over CNN in terms of the adversarial robustness. The authors have conducted comprehensive experiments to support these analyses.

**Limitations And Societal Impact:**

Yes

**Main Review:**

In Line 86, the authors stated that "the adder weights follow Laplace distributions". But I did not get how the authors achieve such a conclusion. Why not the adder weights follow a gaussian distribution? I guess by default, the network is initialized with a gaussian distribution.

What do you mean by "activation xl follows the Rectified Gaussian distribution"?

In Eq. (15), how the authors get the conclusion that the second line has a smaller variance than that of the first line? I could agree that the denominator for ANN is larger than that of CNN, but the numerators are different.

The authors proposed using running mean to improve the adversarial robustness. Will this trick be applied over CNN and what the effect will be?

**Time Spent Reviewing:**

4

---

> ### Author Response · Authors · 2021-08-10
> **Response to Reviewer xDpY**
>
> Thanks for your constructive comments and support.
>
> **Adder Weights Follow Laplace Distribution.**
>
> The weight parameters in ANNs follow Laplace distribution is an observation in both [1] and our empirical studies, as shown in Figure 1 (c). For theoretical analysis, the Laplace distribution in ANNs mainly comes from the fact that the prior of $\ell_1$ norm is Laplace distribution and ANNs make use of $\ell_1$ norm for similarity measurement. Thus, it is natural for ANN weights to follow Laplace distribution after optimization.
>
> **Activation follows Rectified Gaussian Distribution.**
>
> In Eq. 3, we provide one-layer forwarding of ANNs, which includes ReLU function, adder operation and BN layer. The activation $x_l $ denotes the features after ReLU function. Since we assume that the output feature from previous layer follows a Gaussian distribution, the activation after ReLU function follows a Rectified Gaussian Distribution where all negative elements are reset to zero.
>
> **In Eq. (15), how the authors get the conclusion that the second line has a smaller variance than that of the first line? I could agree that the denominator for ANN is larger than that of CNN, but the numerators are different.**
>
> The numerators are different for CNNs and ANNs in Eq. 15. For discussion the numerators, we first remove the influence of $n\sigma_\delta^2$ since they are exactly the same for CNNs and ANNs. The remaining part have similar values where (1-$\frac{2}{\pi}$) is around 0.36 in ANNs while the one in CNNs becomes 1 if we assume the CNN weight follows a standard normal distribution. Although the numerators are similar for both ANNs and CNNs, the denominator of ANN is much larger than the one of CNN, which makes the perturbations in ANNs have a much smaller variance than CNNs.
>
> **Robust inference strategy on CNNs**
>
> In our experiments, CNNs are applied with robust inference strategy for comparison, which uses running mean and tracked variance in BN layers during inference. As shown in Table 1, ‘-R’ denotes this trick, and CNN-R achieves slightly better adversarial robustness than vanilla CNN, which indicates that proposed robust inference strategy also works for CNNs. However, comparing CNN-R with ANN-R, there exist a huge gap, which demonstrates the potential natural robustness of ANNs as discussed in Section 4.

---

### Decision · Program_Chairs · 2021-09-27

**Decision:**

Accept (Poster)

**Comment:**

This paper focuses on enhancing the stability of AdderNets for performance improvement in downstream tasks. The proposal is the adaptive weight normalization (AWN) to optimize adder weight distributions adaptively. The philosophy behind sounds quite interesting to me, namely, batch normalization layers can form automatic perturbation eliminations. This philosophy leads to a simple yet effective inference algorithm design I have never seen.

The clarity and novelty are clearly above the bar of NeurIPS. While the reviewers had some concerns on the significance, the authors did a particularly good job in their rebuttal. Thus, all of us have agreed to accept this paper for publication! Please include the additional experimental results and merge the reviewers' comments in the next version.